# Lifestyle in Undergraduate Students and Demographically Matched Controls during the COVID-19 Pandemic in Spain

**DOI:** 10.3390/ijerph18158133

**Published:** 2021-07-31

**Authors:** María Giner-Murillo, Beatriz Atienza-Carbonell, Jose Cervera-Martínez, Teresa Bobes-Bascarán, Benedicto Crespo-Facorro, Raquel B. De Boni, Cristina Esteban, María Paz García-Portilla, Susana Gomes-da-Costa, Ana González-Pinto, María José Jaén-Moreno, Flavio Kapczinski, Alberto Ponce-Mora, Fernando Sarramea, Rafael Tabarés-Seisdedos, Eduard Vieta, Iñaki Zorrilla, Vicent Balanzá-Martínez

**Affiliations:** 1Department of Medicine, University of Valencia, 46010 Valencia, Spain; magimu912@gmail.com (M.G.-M.); beatrizatica@gmail.com (B.A.-C.); joecerve@hotmail.com (J.C.-M.); 2Hospital de Denia-Marina Salud, 03700 Alicante, Spain; 3Centro de Investigación Biomédica en Red de Salud Mental (CIBERSAM), 28007 Madrid, Spain; mtbobes@gmail.com (T.B.-B.); benedicto.crespo.sspa@juntadeandalucia.es (B.C.-F.); albert@uniovi.es (M.P.G.-P.); anapinto@telefonica.net (A.G.-P.); fscferro69@gmail.com (F.S.); rafael.tabares@uv.es (R.T.-S.); EVIETA@clinic.cat (E.V.); i.zorrilla@gmail.com (I.Z.); 4Servicio de Salud del Principado de Asturias (SESPA), Instituto de Investigación Sanitaria del Principado de Asturias (ISPA), 33011 Oviedo, Spain; 5Department of Psychology, University of Oviedo, 33003 Oviedo, Spain; 6Hospital Universitario Virgen del Rocío, IBIS, 41013 Seville, Spain; cristina.esteban.sspa@juntadeandalucia.es; 7Department of Psychiatry, Faculty of Medicine, University of Sevilla, 41004 Seville, Spain; 8Institute of Scientific and Technological Communication and Information in Health (ICICT), Oswaldo Cruz Foundation (FIOCRUZ), Rio de Janeiro 21040-360, Brazil; raqueldeboni@gmail.com; 9Department of Psychiatry, School of Medicine, University of Oviedo, 33003 Oviedo, Spain; 10Hospital Clinic, Institute of Neuroscience, University of Barcelona, IDIBAPS, Barcelona, 08036 Catalonia, Spain; GOMES@clinic.cat; 11Hospital Universitario de Alava, BIOARABA, UPV-EHU, 01005 Vitoria, Spain; 12Instituto Maimónides de Investigación Biomédica de Córdoba (IMIBIC), 14004 Córdoba, Spain; mjjaen@uco.es; 13Departamento de Ciencias Morfológicas y Sociosanitarias, Universidad de Córdoba, 14071 Córdoba, Spain; 14Department of Psychiatry and Behavioural Neurosciences, McMaster University, Mood Disorders Program, St. Joseph’s Healthcare Hamilton, Hamilton, ON L8S 4L8, Canada; flavio.kapczinski@gmail.com; 15Department of Psychiatry, Universidade Federal do Rio Grande do Sul (UFRGS), Instituto Nacional de Ciência e Tecnologia Translacional em Medicina (INCT-TM), Porto Alegre 90040-060, Brazil; 16CMT-Motores Térmicos, Universitat Politècnica de València, 46002 Valencia, Spain; alponmo@mot.upv.es; 17Hospital Universitario Reina Sofía, Universidad de Córdoba, 14004 Córdoba, Spain; 18Teaching Unit of Psychiatry and Psychological Medicine, Department of Medicine, University of Valencia, 46010 Valencia, Spain

**Keywords:** lifestyle, undergraduate students, mental health, pandemic, COVID-19

## Abstract

Few studies have used a multidimensional approach to describe lifestyle changes among undergraduate students during the COVID-19 pandemic or have included controls. This study aimed to evaluate lifestyle behaviors and mental health of undergraduate students and compare them with an age and sex-matched control group. A cross-sectional web survey using snowball sampling was conducted several months after the beginning of COVID-19 pandemic in Spain. A sample of 221 students was recruited. The main outcome was the total SMILE-C score. Students showed a better SMILE-C score than controls (79.8 + 8.1 vs. 77.2 + 8.3; *p* < 0.001), although these differences disappeared after controlling for covariates. While groups did not differ in the screenings of depression and alcohol abuse, students reported lower rates of anxiety (28.5% vs. 37.1%; *p* = 0.042). A lower number of cohabitants, poorer self-perceived health and positive screening for depression and anxiety, or for depression only were independently associated (*p* < 0.05) with unhealthier lifestyles in both groups. History of mental illness and financial difficulties were predictors of unhealthier lifestyles for students, whereas totally/moderate changes in substance abuse and stress management (*p* < 0.05) were predictors for the members of the control group. Several months after the pandemic, undergraduate students and other young adults had similar lifestyles.

## 1. Introduction

The 2019 coronavirus disease (COVID-19) pandemic is causing an unprecedented global crisis with respect to health and social factors. The pandemic and the measures adopted to combat it have had a substantial impact on the lifestyle habits of people worldwide [1,2].

Lifestyle is currently considered as a multidimensional construct encompassing a set of personal behaviors such as diet/nutrition, substance use, physical activity/exercise, stress management, restorative sleep, social support, and environmental exposures such as screen time and exposure to nature [3]. Healthy lifestyle habits have been consistently shown to influence individuals’ physical health, mental health, and well-being [4,5,6].

An individual’s lifestyle develops throughout life, but late adolescence and early youth are relevant stages in acquiring healthy lifestyle habits. Conversely, this is also a vulnerable period for substance misuse and worsened diet quality, especially when exercise is no longer linked to academic activities. Other notable changes are the increase in the use of screens, the existence of feelings of loneliness and social isolation, and sleep disturbances caused by maturational and psychosocial changes [7]. During an individual’s undergraduate education, students acquire lifestyle behaviors that, in the majority of cases, are maintained in adulthood [8]. Hence, this population represents a key target for health promotion and prevention activities and is the focus of the present study.

During the first semester of 2020, especially during population lockdowns, multiple studies have shown the remarkable impact of COVID-19 pandemic on most lifestyle behaviors of undergraduate students. Thus, significant rates of sleep disturbances have been observed [9,10], as well as decreased physical activity levels and increased sedentary behaviors [9,11,12,13]. A worsening of nutritional and diet quality was also described [14], although a greater adherence to the Mediterranean diet was observed [15] during confinement. Changes have also been demonstrated with respect to substance use [16], such as increased alcohol consumption. Increased use of digital technologies and screen time have also been shown [9], especially among students with psychiatric morbidity and low levels of exercise [17]. Moreover, significant rates of depression, anxiety, and psychological stress have been described among undergraduate students during the pandemic [18,19]. Interestingly, worsened mental health has been related with lifestyle changes in this population [20].

In summary, several studies have described lifestyle changes in undergraduate students during the current pandemic. However, most research so far has focused on some specific areas of healthy lifestyles separately or have used ad hoc, non-validated instruments. In this sense, the Short Multidimensional Inventory Lifestyle Evaluation (SMILE-C) questionnaire was developed and validated during the current pandemic to simultaneously evaluate different dimensions of lifestyle [21]. Moreover, while previous research has been conducted during the first months of the pandemic, longer-term effects have received much less attention. Finally, to our knowledge, no previous studies have included a control group of non-university student participants.

Based on the above, the present study aims to analyze lifestyle changes in undergraduate students, several months after the onset of the pandemic, from a multidimensional perspective (e.g., diet and nutrition, substance use, physical activity, stress management, restorative sleep, social support, and environmental exposures). The main study goal was to describe and compare lifestyles during the survey period between students and a control group of participants matched by age and sex. The secondary objectives were to measure the frequency of mental health problems (depression, anxiety, and alcohol abuse) and to identify independent predictors of healthy lifestyles for each group.

## 2. Materials and Methods

### 2.1. Study Design

This analysis profits from data collected in an online survey conducted from 16 November to 16 December 2020. The online questionnaire was programmed in SurveyGizmo^®^ (http://www.surveygizmo.com.br/ (accessed on 1 November 2020)) and included questions about lifestyle behaviors, and their changes related with the pandemic, demographics, COVID-19 experience, self-rated health, and previous diagnosed conditions, as described elsewhere [21,22]. The survey followed a similar methodology used by our research team during the early pandemic [21,22].

During the survey period, eight months after the declaration of the state of alarm due to the pandemic, restrictions such as home confinement were not issued in Spain. However, the rules for social distancing were reinforced, curfew was established, and the mandatory use of a mask and restriction of mobility between territories were both imposed.

### 2.2. Study Population

The online survey included 3635 adults over 18 years from both sexes living in Spain with internet access who agreed to participate in the study after reading the informed consent form. To avoid the same individuals from answering multiple times, it was asked whether the survey had been answered previously.

For the present study, we selected 221 participants reporting to be undergraduate student, and 221 participants matched by sex and age (range of ± 3 years) who reported not studying or did so in non-university education. The NumPy and Pandas libraries of Python were used, through the Spyder open-source platform for scientific programming, to manipulate and manage the data. The pairing algorithm excludes the already selected individuals to form the subsequent pairs. To avoid repeated responses, it was specifically asked whether the survey had already been answered previously. Thus, the final 221 pairs contain unique individuals. Postgraduate students (e.g., those with a masters or doctorate degree) were excluded from the analyses.

### 2.3. Sampling and Recruitment

The convenience online sample was obtained via social networks (Twitter, Facebook, and WhatsApp), and mailing lists. A snowball technique (i.e., individuals who answered the questionnaire were asked to send the survey link to their contacts) was used. Since the fundamental parameters were unknown at the moment defining sample size, it was not defined a priori. Instead, a 30 day-period of data collection was specified.

### 2.4. Outcome Variable

The main outcome variable was global lifestyle, evaluated using the total score on the SMILE-C [21] scale. This tool was developed from the original SMILE, which is a self-assessed 43-item questionnaire comprising seven lifestyle habits or domains: diet/nutrition, substance abuse, physical activity, stress management, restorative sleep, social support, and environmental exposure. It was developed to carry out a multidimensional and comprehensive assessment of a (healthy) lifestyle during the previous 30 days.

The SMILE-C scale analyses in a global way the lifestyle during the last 30 days, with a questionnaire of 27 items belonging to the seven domains mentioned. The response options have been measured using a 4-point Likert-type scale, the final score being obtained by adding the scores of all the questions (taking into account that some questions have inverse scores). The higher the score, the healthier the lifestyle (scores range from 27 to 108). In a previous publication, it was found that the SMILE-C scale presents a global Cronbach’s alpha of 0.75 and a Kaiser–Meyer–Olkin coefficient = 0.77 [21].

### 2.5. Variables and Measurements

Demographic information included sex, gender, age, local measures to fight against COVID-19 (with options: no measures, perimetral confinement, home confinement), educational level, number of cohabitants at home, and self-isolation by contagion or risk contact. The COVID-19 questions were related to diagnosis (yes/no), need of hospitalization (yes/no) or ventilation (yes/no) and loss of significant ones (yes/no).

Self-rated health (SRH) was measured using the question “How would you rate your health in general?” with possible answer choices of “Very bad”, “Bad”, “Regular”, “Good” and “Very good”. The response options were aggregated into Very good/Good and Regular/Bad/Very bad.

Change in lifestyle behaviors during the COVID-19 pandemic as compared to before the pandemic was assessed by questions such as: “Did you change your (nutritional habits and diet) during the COVID-19 pandemic?”, with a 4-point Likert-type response (Totally, Moderately, Mildly, or Not at all), being aggregated into Totally/Moderately and Mildly/Not at all. To assess whether these changes were towards a more or less healthy lifestyle, we used the question “You consider that your (nutritional habits and diet) nowadays are…”, with three possible answers, “As healthy as before”, “Healthier than before” or “Less healthy than before”.

Previously diagnosed conditions were self-reported using the question “In the last 12 months, have you been diagnosed by a medical doctor or health professional, or received treatment for any of the following conditions?”. Possible health problems investigated included diabetes, heart disease, hypertension, anemia, asthma, depression, anxiety, schizophrenia, bipolar disorder, anorexia/bulimia nervosa, HIV/AIDS, cancer, tuberculosis, cirrhosis, renal disease, and others.

Current depression was screened using the Patient Health Questionnaire-2 (PHQ-2) [23] using a cut-off ≥ 3, and current anxiety was screened using the Generalized Anxiety Disorder 7-item (GAD-7) [24] using a cut-off ≥ 10. Two dichotomous variables were created “Positive Screening for Depression” and “Positive Screening for Anxiety”. Then a composite variable was created using these variables with the following categories: no positive screening, positive screening for depression only, positive screening for anxiety only, and positive screening for both. Screening for alcohol abuse was performed using the Alcohol Use Disorders Identification Test Consumption Questions (AUDIT-C) [25] and cut-off was ≥3.

### 2.6. Statistical Analysis

Following the study methodology of the first survey [21], the mean and standard deviation of the SMILE-C scores were calculated for all the control group and of undergraduate students’ variables. Normality of distribution was tested for all the variables using the Kolmogorov–Smirnov test. Therefore, statistically significant differences were evaluated using non-parametric tests. For the bivariate associations between dichotomous variables and the SMILE-C scores, Mann–Whitney U test was used. The McNemar test was used for lifestyle changes and mental health between the paired groups. The independent variables were described by results and compared proportions using χ2. The bivariate associations between age and the number of people living in the household and the SMILE-C score were evaluated using the Spearman correlation tests.

In order to analyze between-group differences in SMILE-C scores, a multivariate linear regression model was performed to control the potential influence of covariates where the group (undergraduate students/controls) and all variables associated with the SMILE-C with *p* < 0.20 were entered as independent variables. Multivariate linear regression models were also performed to evaluate the effect of independent factors on the total SMILE-C score of each group. Initial models included variables associated with the SMILE-C with *p* < 0.20 in the bivariate analyses. In order to find out the B of each category of the categorical variables, dummy variables (binary) were created for each category of them. A final model was reached for each group, using a manual stepwise removal of each non-significant variable, and evaluating the changes in the remaining B.

### 2.7. Ethical Aspects

The study was approved by the Ethics Committee at the Hospital Universitari i Politècnic La Fe, in Valencia, Spain (2020-149-1). The surveys were anonymous (no identification -name-, city or IP address was collected) and participants read the consent form and confirmed their interest in participating in the first screen of the online questionnaires.

## 3. Results

A sample of 442 matched participants by age and sex was selected for our study (221 undergraduate students and 221 individuals who were not studying or did so in non-university education). Table 1 describe the sociodemographic and clinical characteristics of the sample, COVID-19 related experiences, and the respective SMILE-C means and their bivariate association.

Regarding sociodemographic variables, almost three quarters of the sample (73.3%) was composed of women. Participants were demographically distributed as follows: 38.9% from the Valencian Community, 34.6% from Andalusia, and 26.5% from other regions. The mean age for the undergraduate students was 21.86 ± 2.12 years and 22.86 ± 2.48 years for controls. In both groups, none of the sociodemographic variables were significantly associated with the SMILE-C score (Table 1), although in controls a non-significant trend was observed for the number of people living in the household. In both groups, having experienced financial difficulties during the pandemic was the only variable significantly associated with lower SMILE-C scores (Table 1). No other COVID-19 related variables were associated with the SMILE-C score.

In the control group, higher SMILE-C scores were associated with several health-related variables: very good or good SRH, not being diagnosed or treated in the previous year of mental illness, psychiatric disorders (schizophrenia/bipolar disorder/anorexia/bulimia), or other diseases (HIV/AIDS, tuberculosis, cancer, cirrhosis, other), and having a negative screening for depression and anxiety. In the group of students, the following variables were associated with higher SMILE-C scores: very good or good SRH, no history of mental illness, and a negative screening for depression and anxiety. In contrast, positive screening for alcohol abuse was not associated with SMILE-C total scores in either of the two groups.

### 3.1. Comparison of Lifestyle Behaviours and Mental Health

Undergraduate students presented a higher mean SMILE-C score than controls (79.81 + 8.14 vs. 77.21 + 8.31, *p* < 0.001). However, after controlling the effect of several covariates on the SMILE-C score, these differences disappeared (*p* = 0.162). A regression model including number of cohabitants (*p* = 0.001), self-rated health (*p* < 0.001), screening for anxiety (*p* < 0.001), economic difficulties during the pandemic (*p* < 0.001), and changes in stress management (*p* < 0.001) and in restorative sleep (*p* = 0.006) explained 35.7% of variance in SMILE-C mean scores (F = 34.36, R^2^ = 0.357, *p* < 0.001).

Groups did not differ in the screenings of depression and alcohol abuse, although significant differences were found for the screening of anxiety. Regarding the proportion of relevant changes (totally/moderate) in the seven lifestyle habits, groups only differed in diet/nutrition and restorative sleep. In both cases, the changes were more relevant in controls (Table 2).

### 3.2. Changes on Lifestyle Behaviours during the COVID-19 Pandemic

The self-reported changes in lifestyle behaviors during the COVID-19 pandemic are depicted for both groups (Figure 1 and Figure 2). Table 3 shows the distribution of these changes between both groups. Significant differences were found in environmental exposures only. Compared with the controls, a higher proportion of undergraduate students, maintained their habits of environmental exposures just as healthy as before the pandemic.

### 3.3. Variables Independently Associated with Lifestyle Behaviours

The final multivariate models for both groups are shown on Table 4 and Table 5. For the control group (Table 4), the variables that remained independently associated with higher SMILE-C scores were a higher number of people living in the household and the total/moderate changes in stress management. A worse SRH, a positive screening for depression/anxiety, and total/moderate changes in substance abuse were associated with lower SMILE-C scores.

For students (Table 5), the number of people living in the household was the only variable that remained independently associated with higher SMILE-C scores. In contrast, a worse Self-Rated Health, a positive screening only for depression or depression and anxiety, having a history of mental illness, and having experienced financial issues during the pandemic were associated with lower SMILE-C scores.

## 4. Discussion

In this online survey conducted several months after the lockdown in Spain, overall lifestyle measured by the total score on the SMILE-C scale was healthier among undergraduate students compared to that of sex and age-matched participants from the general population in Spain. However, when controlling for covariates, these differences were no longer significant. Positive screenings for anxiety were significantly higher in the control group and non-significant trends in the same direction were also observed for screenings for depression and alcohol abuse. Moreover, this study further confirms the important association between mental health and healthy lifestyles in a general population sample and expands this finding to undergraduate students.

Both groups showed similar lifestyles during the survey period. The interpretation of this innovative finding is hindered by the absence of previous studies comparing undergraduate students with matched controls. As a more educated group, students are expected to have a higher health literacy [26]. Indeed, a higher literacy on a healthy diet has been reported to be associated with healthier eating behaviors among undergraduate students during the current pandemic [27]. However, according to the present results, that would not drive a major advantage over controls in adherence to a healthier lifestyle. Of note, differences in lifestyles between groups became non-significant once several potential confounders were taken into account. In particular, the number of cohabitants, self-rated health, screening for anxiety, having experienced financial difficulties during the pandemic, and changes in stress management and in restorative sleep were all significantly associated with SMILE-C scores.

Regarding mental health problems, about 30% of the undergraduate students had a positive screening for either depression or anxiety, while alcohol abuse was suspected in 15%. Overall, these rates were lower than those of the control participants in this study, as mentioned above. The COVID-19 pandemic has notoriously affected the mental health of the general population [28,29]. According to a recent meta-analysis [30], the prevalence of anxiety does not seem to have increased during the COVID-19 pandemic in medical students, although increased stress has been found in other undergraduate students [18]. Prior to the pandemic, substantial evidence supports that undergraduate students have higher rates of anxiety than the general population [31]. Taken together, it could be speculated that anxiety likely increased in both groups of this survey, but more so in controls, until they significantly exceeded those of undergraduate students.

It is likely that getting earlier and a higher amount of information about the pandemic, as well as higher resilience developed throughout the grades, can protect students against stress-related symptoms. Conversely, the greater likelihood of experiencing financial difficulties seems to be a risk factor for controls. These hypotheses could also apply to depression and await confirmation with further comparative, ideally longitudinal, studies.

When each lifestyle habit was compared between groups, statistically significant differences arose only in diet/nutrition and restorative sleep. In both cases, students had a lower percentage of totally/moderate changes than the controls. Despite having taken online lessons on many occasions, students may have also continued to maintain a regular schedule of chores over the pandemic, which in turn may account for these findings.

In undergraduate students, physical activity was the lifestyle behavior most sensitive to the recent effects of the pandemic, since almost half of the sample reported totally/moderate relevant changes during the previous month. These results converge with those of previous studies that analyzed changes in physical activity [9,11,12,32]. Moreover, physical activity is among the most common strategies to help with mental wellbeing [33], which further emphasizes the link between lifestyles and common mental symptoms [34]. Lagging behind physical activity, one third of students reported having experienced relevant changes over the previous month in stress management, environmental exposures, and social support. Previous surveys suggest that stress was higher among respondents with sleep problems [35], substance use [20], alcohol-related problems, eating problems, and problematic internet use [36]. Moreover, those spending at least two hours outside, or less than eight hours engaged on electronic screens, were likely to experience lower levels of psychological impact [37]. Our results are also consistent with those of other studies [9,15], which found that screen time has increased substantially during the pandemic. Moreover, a lower perception of social support was associated with increased depressive symptoms in undergraduate students during the third wave in Hong Kong [38].

As another remarkable finding of this survey, lifestyles of all of the students remained mostly unchanged compared to those before the pandemic. Regarding substance use, more than a third of participants claimed to have healthier habits than before the pandemic. This is an expected finding given the substantial pandemic-related decrease in social activities that could worsen overall lifestyle. These results are also consistent with reduced alcohol use previously reported in a university sample [39].

Changes compared to before the pandemic observed in students significantly differed from those of controls for environmental exposures only. Before the COVID-19, students were already used to spending a large amount of time with screens as part of their learning chores. Although a longer screen time has been observed during the pandemic, this increase has been lower than that in non-university participants.

Despite the fact that both groups showed similar lifestyles, several variables were independently associated with healthier or unhealthier lifestyles. For both groups, self-perceiving a good state of health was strongly associated with a healthier lifestyle. We believe that this result gives value to this questionnaire since the participants who report having a worse state of health are in turn those who obtain a lower total score on the SMILE-C (i.e., an unhealthier overall lifestyle). Secondly, a greater number of cohabitants was also associated with healthier lifestyles in both groups. Related with this, the restriction of social interaction outside the home could improve the relationships with cohabitants, which in turn may favour strengthening social support. Moreover, living with more people could have favored the organization of daily routines, and even the collaborative improvement of lifestyle such as sharing meals and exercise routines. Previous studies have shown gender differences in attitudes and behaviors [40,41], as well as several mental health outcomes [42] during the COVID-19 pandemic. However, in this study no significant gender differences were observed in either group regarding overall lifestyles.

In undergraduate students, the risk for a less healthy lifestyle was higher in those with a history of mental illness in the previous year or a probable depression with or without anxiety, compared with those with a negative screening for both common mental disorders. Several studies have found that poor mental health is associated with lifestyle in undergraduate students. For instance, a relationship between anxiety symptoms and sleep problems has been described [43], whereas engaging in physical activity [44] and a higher perceived social support [20,38] reduced the likelihood of anxiety. Moreover, depressive symptoms seem to be more prevalent among students with sleep problems [43], higher screen time [38,45], decreased physical activity [46,47], poor social support [20], and smoking or alcohol consumption [33]. The present results are also consistent with previous evidence showing that psychiatric disorders, in general, are associated with unhealthy lifestyle behaviors [48]. In addition, a close relationship was observed between unhealthier lifestyles and having experienced economic difficulties during the pandemic, as well as with the preference for not answering this question. This is an expected result since a lower purchasing power is usually associated with poorer lifestyles.

The interpretation of the present findings must be understood in the context of several limitations. Firstly, the external validity of the results is constrained by the representativeness of the sample. At the moment, web surveys are still considered non-probability samples, as the probability of inclusion of each individual is unknow and individuals without access to the web have a null chance to be included. Thus, this sample is not representative neither from the Spanish general population nor the Spanish undergraduate students. As in other web surveys performed during the pandemic, women were overrepresented in our sample (73%) [9,11,12,49,50] as compared to the percentage of women in the university (55%) [51] and in the Spanish population (approximately 50%) [52]. In addition, only 37.5% [53] of the Spanish population has an undergraduate degree and our results must be interpreted with caution.

Secondly, the analysis of lifestyle changes was based on self-reported perception, which is subject to social desirability and memory bias. New technologies, such as personal health trackers, can be helpful in overcoming these limitations in the future. Thirdly, the SMILE-C was designed to use a 4-point Likert-type scale (always, often, seldom, never). This might be seen as a limitation since it may force respondents to not give neutral answers. Whether or not offering a central or middle point in a Likert-type scale has been disputed for decades. The original meaning of the midpoint is neutral or indifferent, for instance ‘neither agree nor disagree’. Likert scales with an uneven number of options, usually five or seven, are the standard to measure degree of (dis)agreement. Nevertheless, the SMILE-C measures the frequency of a given lifestyle behavior, instead. Moreover, this is a complex issue. Although Likert scales with four or seven points have been found to have the strongest support in terms of reliability and validity [54], less research has examined changes to the characteristics of the data when a midpoint is utilized [55]. In addition, Garland [56] found that social desirability bias can be minimized by eliminating the midpoint. Lastly, like in any cross-sectional study, reverse causality cannot be excluded, and associations must be interpreted with caution.

This study has several strengths. Most similar studies have examined aspects of lifestyle in isolation, such as diet/nutrition, restorative sleep, or stress, whereas the simultaneous assessment of more than three habits is scarce. Our study is among the first to evaluate the lifestyles of university students from a multidimensional approach. Moreover, most previous research has focused on the first months of the pandemic, during home confinement. The evaluation of healthy lifestyle eight to nine months after the lockdown in Spain allows us to better understand the long-term effects of the pandemic. Finally, to our knowledge, this report is the first to include not only a comparison group but also age and sex-matched participants to analyze the lifestyles of undergraduate students during the current pandemic. This design allows establishing whether the observed changes are specific to that population group or are more generic. The importance of taking into account confounding variables, such as anxiety or having experienced financial difficulties during the pandemic, should be emphasized in order to better understand the likely complex pattern of lifestyle determinants and to obtain more valuable and accurate information on lifestyle changes in undergraduate students.

## 5. Conclusions

The present study showed that participants had remarkable changes in lifestyle several months after the onset of the COVID-19 pandemic in Spain. A series of demographic and clinical variables were found to be independently associated with lifestyle during the study period. Consequently, more attention should be paid to changes in lifestyle behaviors of the young adult population during health crises, such as the current pandemic, in search of protective factors and markers of resilience. The results of this and future studies can be used to refine the public health recommendations issued to maintain or adopt healthier lifestyles.

## Figures and Tables

**Figure 1 ijerph-18-08133-f001:**
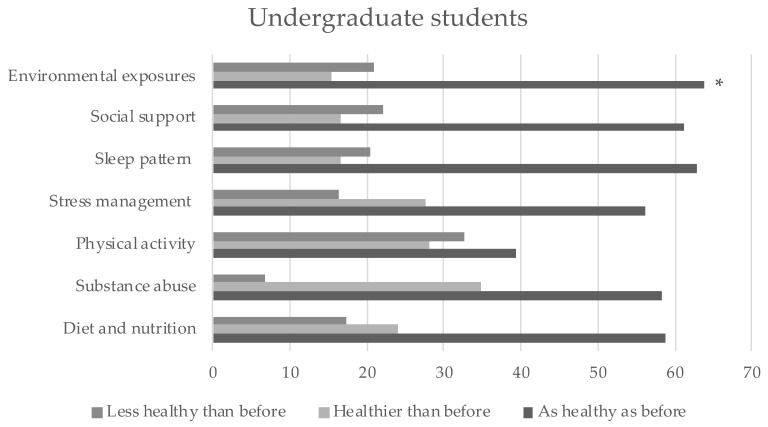
Self-reported changes on lifestyle behaviors during the COVID-19 pandemic in undergraduate students. * Significant differences (*p* = 0.004).

**Figure 2 ijerph-18-08133-f002:**
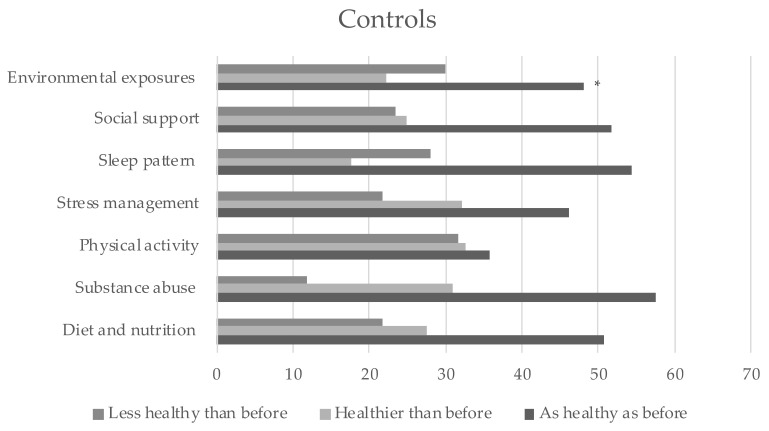
Self-reported changes on lifestyle behaviors during the COVID-19 pandemic in controls. * Significant differences (*p* = 0.004).

**Table 1 ijerph-18-08133-t001:** Sociodemographic, COVID-19 related experiences, health-related variables and mean SMILE-C score.

Variables	Controls			Undergraduate Students		
	n (%)	SMILE-C Mean (SD)/r	*p*-Value	n (%)	SMILE-C Mean (SD)/r	*p*-Value
Sex						
Male	59 (26.70)	76.85 (7.25)	0.449	59 (26.70)	80.95 (9.33)	0.086
Female	162 (73.30)	77.34 (8.68)		162 (73.30)	79.4 (7.65)	
Age *	23 (21–25)	r = 0.076	0.259	22 (21–23)	r = −0.002	0.979
Number of cohabitants *	3.0 (2.5–4.0)	r = 0.129	0.056	4.0 (3.0–4.0)	r = 0.108	0.109
COVID−19 diagnosis						
No	203 (91.90)	76.99 (8.32)	0.101	209 (94.57)	79.58 (8.13)	0.132
Yes	18 (8.10)	79.67 (8.06)		12 (5.43)	83.83 (7.47)	
Lost somebody in the pandemic						
No	204 (92.30)	77.25 (8.14)	0.591	202 (91.40)	79.8 (8.02)	0.817
Yes	17 (7.70)	76.64 (10.40)		19 (8.60)	80.0 (9.57)	
Economic difficulties during the pandemic						
No	151 (68.30)	79.09 (6.93)	<0.001	179 (81.00)	80.88 (7.36)	0.005
Yes	58 (26.20)	73.74 (8.89)		28 (12.67)	75.18 (8.33)	
Prefer not to answer	12 (5.40)	70.33 (12.44)		14 (6.33)	75.5 (12.43)	
Self-rated health						
Very good or good	176 (79.64)	79.19 (6.78)	<0.001	192 (86.88)	81.12 (7.16)	<0.001
Regular, bad or very bad	45 (20.36)	69.47 (9.24)		29 (13.12)	71.17 (9.03)	
Diagnosed or treated for mental illness during the last year						
No	176 (79.64)	78.36 (7.74)	<0.001	175 (79.18)	81.01 (7.53)	<0.001
Yes	45 (20.36)	72.69 (8.99)		46 (20.81)	75.28 (8.86)	
Screening for depression and anxiety						
Negative for both depression and anxiety	123 (55.66)	80.17 (6.90)	<0.001	142 (64.25)	82.94 (6.90)	<0.001
Positive for depression only	16 (7.24)	75.12 (6.56)		16 (7.24)	74.18 (6.36)	
Positive for anxiety only	30 (13.57)	75.43 (5.97)		15 (6.79)	78.6 (6.43)	
Positive for both depression and anxiety	52 (23.53)	71.86 (9.89)		48 (21.72)	72.83 (7.20)	
Screening for alcohol abuse						
Negative	184 (83.26)	77.47 (8.11)	0.541	188 (85.07)	79.79 (8.25)	0.978
Positive	37 (16.74)	75.92 (9.27)		33 (14.93)	79.94 (7.59)	
Diagnosed or treated for schizophrenia/bipolar disorder/anorexia/bulimia in the previous year						
No	214 (96.83)	77.51 (7.97)	0.036	216 (97.73)	79.82 (8.16)	0.977
Yes	7 (3.17)	68.0 (13.19)		5 (2.62)	79.40 (8.08)	
Diagnosed or treated for diabetes in the previous year						
No	212 (95.93)	77.39 (8.31)	0.088	217 (98.20)	79.83 (8.10)	0.893
Yes	9 (4.07)	72.89 (7.47)		4 (1.80)	78.75 (11.44)	
Diagnosed or treated for asthma/bronchitis in the previous year						
No	205 (92.76)	77.26 (8.42)	0.483	207 (93.70)	79.77 (8.23)	0.836
Yes	16 (7.24)	76.56 (7.01)		14 (6.30)	80.50 (6.97)	
Diagnosed or treated for heart disease or hypertension in the previous year						
No	211 (95.47)	77.21 (8.41)	0.929	218 (98.60)	79.79 (8.15)	0.585
Yes	10 (4.53)	77.20 (6.30)		3 (1.40)	81.67 (9.29)	
Diagnosed or treated for chronic disease in the previous year						
No	179 (81.00)	77.61 (8.62)	0.044	185 (83.71)	79.87 (8.17)	0.977
Yes	42 (19.00)	75.5 (6.66)		36 (16.29)	79.53 (8.10)	
Diagnosed or treated for others (HIV/AIDS, tuberculosis, cancer, cirrhosis, kidney disease, other)						
No	192 (86.88)	77.69 (7.91)	0.061	192 (86.88)	79.98 (7.84)	0.669
Yes	29 (13.12)	74.00 (10.15)		29 (13.12)	78.68 (9.98)	

SMILE-C: Short Multidimensional Inventory Lifestyle Evaluation, SD: Standard Deviation, * Median (Interquartile range)

**Table 2 ijerph-18-08133-t002:** Between-group comparison of mental health screenings and lifestyle changes.

Variables	Controls n (%)	Undergraduate Students n (%)	*p*-Value
Screening of alcohol abuse			
Negative	184 (83.3)	188 (85.1)	0.093
Positive	37 (16.7)	33 (14.9)	
Screening of depression			
Negative	153 (69.2)	157 (71.0)	0.077
Positive	68 (30.8)	64 (29.0)	
Screening of anxiety			
Negative	139 (62.9)	158 (71.5)	0.042
Positive	82 (37.1)	63 (28.5)	
Lifestyle changes:			
Diet and nutrition			
Mild/no changes	156 (70.6)	178 (80.5)	0.018
Totally/moderate changes	65 (29.4)	43 (19.5)	
Substance abuse			
Mild/no changes	178 (80.5)	189 (85.5)	0.228
Totally/moderate changes	43 (19.5)	32 (14.5)	
Physical activity			
Mild/no changes	112 (50.7)	120 (54.3)	0.497
Totally/moderate changes	109 (49.3)	101 (45.7)	
Stress management			
Mild/no changes	139 (62.9)	154 (69.7)	0.137
Totally/moderate changes	82 (37.1)	67 (30.3)	
Restorative sleep			
Mild/no changes	153 (69.2)	176 (79.6)	0.022
Totally/moderate changes	68 (30.8)	45 (20.4)	
Social support			
Mild/no changes	144 (65.2)	161 (72.9)	0.111
Totally/moderate changes	77 (34.8)	60 (27.1)	
Environmental exposures			
Mild/no changes	137 (62.0)	156 (70.6)	0.078
Totally/moderate changes	84 (38.0)	65 (29.4)	

**Table 3 ijerph-18-08133-t003:** Self-reported changes on lifestyle behaviors compared with before the pandemic.

Variables	Controls n (%)	Undergraduate Students n (%)	*p*-Value
Diet and nutrition			
As healthy as before	112 (50.7)	130 (58.8)	0.216
Healthier than before	61 (27.6)	53 (24.0)	
Less healthy than before	48 (21.7)	38 (17.2)	
Substance abuse			
As healthy as before	127 (57.5)	129 (58.4)	0.172
Healthier than before	68 (30.8)	77 (34.8)	
Less healthy than before	26 (11.8)	15 (6.8)	
Physical activity			
As healthy as before	79 (35.7)	87 (39.4)	0.56
Healthier than before	72 (32.6)	62 (28.1)	
Less healthy than before	70 (31.7)	72 (32.6)	
Stress management			
As healthy as before	102 (46.2)	124 (56.1)	0.1
Healthier than before	71 (32.1)	61 (27.6)	
Less healthy than before	48 (21.7)	36 (16.3)	
Restorative sleep			
As healthy as before	120 (54.3)	139 (62.9)	0.126
Healthier than before	39 (17.6)	37 (16.7)	
Less healthy than before	62 (28.1)	45 (20.4)	
Social support			
As healthy as before	114 (51.6)	135 (61.1)	0.068
Healthier than before	55 (24.9)	37 (16.7)	
Less healthy than before	52 (23.5)	49 (22.2)	
Environmental exposures			
As healthy as before	106 (48.0)	141 (63.8)	0.004
Healthier than before	49 (22.2)	34 (15.4)	
Less healthy than before	66 (29.9)	46 (20.8)	

**Table 4 ijerph-18-08133-t004:** Factors associated with better SMILE-C scores in the control group.

Variables	Controls	
	B CI (95%)	*p*-Value
Number of cohabitants	1.057 (0.195–1.919)	0.017
Self-rated health		
Very good or good	Reference	
Regular, poor or very poor	−7.674 (−10.045–5.304)	<0.001
Screening for depression and anxiety		
Negative for both anxiety and depression	Reference	
Positive for depression only	−3.958 (−7.531–0.385)	0.03
Positive for anxiety only	−3.679 (−6.425–0.934)	0.009
Positive for both depression and anxiety	−5.626 (−8.002–3.25)	<0.001
Substance abuse		
Mild/no changes	Reference	
Totally/moderate changes	−2.422 (−4.759–0.085)	0.042
Stress management		
Mild/no changes	Reference	
Totally/moderate changes	3.010 (1.140–4.881)	0.002

**Table 5 ijerph-18-08133-t005:** Factors associated with better SMILE-C score among the undergraduate students group.

Variables	Undergraduate Students	
	B CI (95%)	*p*-Value
Number of cohabitants	1.101 (0.249–1.953)	0.012
Self-rated health		
Very good or Good	Reference	
Regular, bad or very bad	−5.729 (−8.494–2.964)	<0.001
Screening for depression and anxiety		
Negative for both anxiety and depression	Reference	
Positive for depression only	−7.575 (−10.921–4.230)	<0.001
Positive for anxiety only	−2.890 (−6.382–0.603)	0.104
Positive for both depression and anxiety	−7.341 (−9.677–5.006)	<0.001
Diagnosed or treated for a mental illness in the previous year		
No	Reference	
Yes	−2.379 (−4.632–0.127)	0.039
Economic difficulties during the pandemic		
No	Reference	
Yes	−2.665 (−5.325–0.004)	0.050
Prefer not to respond	−3.932 (−7.485–0.379)	0.030

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
