# Peer review of "Lifestyle in Undergraduate Students and Demographically Matched Controls during the COVID-19 Pandemic in Spain"

_ijerph, 2021, doi:10.3390/ijerph18158133_

Round 1
Reviewer 1 Report
The manuscript entitled „Lifestyle in undergraduate students and demographically matched controls during the COViD-19 pandemic in Spain” presents interesting issues but the manuscript should be corrected.
Major:
- Authors justified their study as novel approach comparing the lifestyle of undergraduate students and demographically matched controls (not being undergraduate students), but it is still not explained why the undergraduate students are such important group to be studied and compared with the matched controls, so the study is still not justified.
- Authors di not assess representativeness of their studied group, so in fact we do not know what is the value of the obtained results.
- The recruited control sample differed significantly from the studied group taking into account their economic difficulties, health and depression, so the differences of lifestyle may have been supposed, Moreover, we do not know what is the reason of observed differences – is it the under graduation, of indicated above other differences. As a result, we can not conclude about it, however, information about representativeness of both studied group would facilitate comparison.
- Authors used unusual version of the Likert scale, as the typical one should be 5-poins or 7-points (preferable, as it allows deepen analysis), and 4-points scale is used only as a ‘forced’ scale to force respondent to not give neutral answers. At the same time, Authors used a 4-points scale but with the neutral answer. Such approach must be properly justified, as currently 7-points scale is recommended, so reducing points to 4 only reduces the value of the manuscript).
Editing:
The manuscript is shabbily prepared and should be corrected (e.g. missing spaces or redundant spaces, tables not prepared according to instructions for authors, etc.).
Abstract:
Authors should present specific results of their study accompanied by results of their statistical analysis.
Introduction:
Authors should be more focused on presentation of the real justification of the presented study. In the current version of this Section there is no real justification of why the undergraduate students are such important group to be studied.
Authors should not include excessive number of references to this section, as Authors should not present all the studies which they found, but only crucial ones. Instead on small studies, they should rather include big international, or big national studies (not only those conducted by their team or in their country), especially those published in a prominent journals.
Materials and Methods:
Authors should present specific information about representativeness of the studied groups while compared with the general population.
It seems that Authors did not verify the normality of distribution and they treated all the variables as normally distributed.
Authors should (1) verify the normality of distribution, (2) for normally distributed data present mean and SD values, but for the other distributions – present median, min and max values, (3) apply adequate statistical tests, that are based on the distribution.
Results:
Authors should verify normality of distribution.
For normally distributed data Authors should present mean and SD values, but for the other distributions – present median, min and max values.
Authors should apply adequate statistical tests, that are based on the distribution.
Data presented in tables should not be extensively reproduced in the tables.
Discussion:
Authors should include the issue of gender dependent behaviours and approaches in populations during COVID-19 pandemic, e.g.: https://pubmed.ncbi.nlm.nih.gov/33060298/; https://www.ncbi.nlm.nih.gov/pmc/articles/PMC7459707/; https://www.ncbi.nlm.nih.gov/pmc/articles/PMC7603590/.
Authors should extensively discuss the limitations of their study (see above).
Conclusions:
Authors should not have references in this section.
Authors Contributions:
It seems that majority of Authors did not present in the manuscript preparation. It is a serious risk of the guest authorship procedure, which is forbidden. They should be either not included as authors (and presented only in Acknowledgements section), or their participation in manuscript preparation should be indicated.
References:
Authors should reduce the number of self-citations, as based on their References section one could think that they are only authors who studied lifestyle changes during COVID. Instead of excessive number of their own citations (ref. 1, 2, 38, 52, 53), especially those published in their national journals, and instead they should present international approach and include publications from various countries published in more prominent international journals.
Reviewer 2 Report
Thanks for allowing me to review the paper entitled “Lifestyle in undergraduate students and demographically matched controls during the COViD-19 pandemic in Spain”
Abstract :
Well, done. We can observe a structured abstract. Will me necessary as well to specify in the abstract how the sample was realized.
Keywords: I suggest removing "Online Survey" as a Keyword due to it is not a descriptor in Health Sciences. If you are using MESH as keywords, I suggest changing it into another more appropriate one.
Introduction. Good. Nothing remarkable.
Material and Methods
- Study Design. We accept the cross-sectional and multicentric study but the center's participants should be described ( %) The online questionnaire was programmed with Survey Gizmon but must be specified if this program has IP filtering or how you control a survey is not done twice.? In the ethical aspects IP filtering is excluded, so please explain.
- Study Population and Sample: Please indicate the centers of the 3635 patients, their distribution around Spain ( at least autonomous communities) In addition specify inclusion and exclusion criteria. Please define % of studies :
a. Recruitment. In addition, an important lack of the study is determinate how filtering was established, again how is possible to determinate :
b. A student does not do the survey, once, twice, or multiple times??
c. In addition, how are known students less than 18 years old are excluded? Is it possible to chose age as a Likert scale.? These parameters should be more clearly explain.
Results.
Please clearly specify who is the control group.
Please specify how is measure the screening for depression, anxiety, alcohol abuse ….
Discussion. Correctly.
Round 2
Reviewer 1 Report
The manuscript entitled „Lifestyle in undergraduate students and demographically matched controls during the COViD-19 pandemic in Spain” presents interesting issues but the manuscript should be corrected. Unfortunately, Authors ignored a number of my comments from my previous review.
Major:
- Authors did not assess representativeness of their studied group, so in fact we do not know what is the value of the obtained results. Authors should present specific information about representativeness of the studied groups while compared with the general population. Authors should take general information about their national population (at least gender in the age groups) and compare with the data for their studied populations.
- The recruited control sample differed significantly from the studied group taking into account their economic difficulties, health and depression, so the differences of lifestyle may have been supposed, Moreover, we do not know what is the reason of observed differences – is it the under graduation, of indicated above other differences. As a result, we can not conclude about it, however, information about representativeness of both studied group would facilitate comparison. It should be presented as a limitation of the study.
- Authors used unusual version of the Likert scale, as the typical one should be 5-poins or 7-points (preferable, as it allows deepen analysis), and 4-points scale is used only as a ‘forced’ scale to force respondent to not give neutral answers. At the same time, Authors used a 4-points scale but with the neutral answer. Such approach must be properly justified, as currently 7-points scale is recommended, so reducing points to 4 only reduces the value of the manuscript). Authors should answer this issue in their manuscript, to provide rationale for readers, not only in the response letter.
Abstract:
Authors should present specific results of their study (not only the results of their statistical analysis).
Introduction:
Authors should not include excessive number of references to this section, as Authors should not present all the studies which they found, but only crucial ones. Instead on small studies, they should rather include big international, or big national studies (not only those conducted by their team or in their country), especially those published in a prominent journals.
Materials and Methods:
Authors should present specific information about representativeness of the studied groups while compared with the general population.
Authors should verify normality of distribution and indicate the test which was used.
In spite of the fact that for the population distribution of variables will approximate a normal distribution, we can not treat non parametric distributions as parametric ones and Authors must apply approach based on a real distribution observed for their sample.
Results:
Authors should apply adequate statistical tests, that are based on the distribution.
Discussion:
Authors should extensively discuss the limitations of their study (see above – the other issues should be included also).
References:
Authors should reduce the number of self-citations, as based on their References section one could think that they are only authors who studied lifestyle changes during COVID. Instead of excessive number of their own citations, especially those published in their national journals, and instead they should present international approach and include publications from various countries published in more prominent international journals.
Reviewer 2 Report
Dear Authors. Thanks so much for your feedback and for improving the article, which is good.
However, still for me is not clear the sampling procedures.
Thanks so much for sending the reference to Lancet Study, is true, that the Snowball sampling process is good for a qualitative study, like the study you refer to: Liu, Q., Luo, D., Haase, J. E., Guo, Q., Wang, X. Q., Liu, S., ... & Yang, B. X. (2020). The experiences of health-care providers during the COVID-19 crisis in China: a qualitative study. The Lancet Global Health, 8(6), e790-e798.
However in this study is not completely adequate the sampling procedures, at least you should justify it. You recollected variables categorized by which regression is done !
Please clarify it - unless you are doing a mixed-method study- , still is not clear.
The rest of the article has been improved, so thanks so much. and congratulations.
